# Various Endoscopic Techniques for Treatment of Consequences of Acute Necrotizing Pancreatitis: Practical Updates for the Endoscopist

**DOI:** 10.3390/jcm9010117

**Published:** 2020-01-01

**Authors:** Mateusz Jagielski, Marian Smoczyński, Jacek Szeliga, Krystian Adrych, Marek Jackowski

**Affiliations:** 1Department of General, Gastroenterological and Oncological Surgery, Collegium Medicum Nicolaus Copernicus University, 87-100 Toruń, Poland; jacky2@wp.pl (J.S.); jackowscy@homail.com (M.J.); 2Department of Gastroenterology and Hepatology, Medical University of Gdańsk, 80-214 Gdańsk, Poland; kgastro@gumed.edu.pl (M.S.); krystian@gumed.edu.pl (K.A.)

**Keywords:** acute necrotizing pancreatitis, pancreatic necrosis, endotherapy, therapeutic endoscopy, therapeutic EUS, minimally invasive

## Abstract

Despite great progress in acute pancreatitis (AP) treatment over the last 30 years, treatment of the consequences of acute necrotizing pancreatitis (ANP) remains controversial. While numerous reports on minimally invasive treatment of the consequences of ANP have been published, several aspects of interventional treatment, particularly endoscopy, are still unclear. In this article, we attempt to discuss these aspects and summarize the current knowledge on endoscopic therapy for pancreatic necrosis. Endotherapy has been shown to be a safe and effective minimally invasive treatment modality in patients with consequences of ANP. The evolution of endoscopic techniques has made endoscopic drainage more effective and reduced the use of other minimally invasive therapies for pancreatic necrosis.

## 1. Introduction

According to the 2012 revised Atlanta classification of acute pancreatitis (AP), two main pathological forms of AP can be distinguished: The edematous and necrotizing forms [1,2,3,4]. In the majority of cases (80–90%), the edematous form, which is characterized by generalized inflammation with edema, prevails [3,4]. Acute necrotizing pancreatitis (ANP) is observed in 10–20% of cases, and is diagnosed by the presence of pancreatic parenchyma and/or surrounding tissue necrosis [3,4]. Mixed necrosis (including parenchyma and surrounding tissues) is observed in 75–80% of cases of ANP [5]. Far less common is peripheral necrosis (involving only the surrounding tissues without the parenchyma—20% of cases) or central necrosis (parenchymal necrosis without surrounding tissue involvement—5% of cases) [5].

In the course of AP, local complications, such as pancreatic and peripancreatic fluid collections, can arise. Depending on the disease stage and morphology, four types of fluid collections can be distinguished: Acute peripancreatic fluid collection (APFC), pancreatic pseudocyst (PPC), acute necrotic collection (ANC), and walled-off pancreatic necrosis (WOPN) [1,2,3,4]. Each fluid collection can be either sterile or infected [1,2,3,4].

In ANP, pancreatic fluid collections (PFCs) developing within the first four weeks are referred to as ANC [1,2,3,4]. However, after four weeks, the remaining collections are termed WOPN [1,2,3,4]. ANC (Figure 1a,b) is an ill-bordered fluid reservoir rich in necrotic tissues and developing within the first four weeks in most ANP patients [3,4,6,7]. Almost half of ANCs will relapse, while the other half will evolve into WOPN [6,7]. WOPN (Figure 2a,b) is a well-bordered PFC, which develops after four weeks of ANP and contains varying amount of liquefied necrotic tissues, depending on time since ANP onset [1,2,3,4,6,7].

## 2. Strategy of Interventional Treatment

The main indication for interventional treatment of consequences of ANP are infected pancreatic and peripancreatic fluid collections [8,9,10,11,12]. Interventional treatment is also required in patients with clinical symptoms directly associated with the collections, such as compression symptoms (mechanical jaundice, ileus, etc.) [8,10,11,12]. Patients with asymptomatic PFC, regardless of the size, do not require interventions [8,10,11,12].

Over the last three decades, the management of PFCs due to ANP has improved as a result of better medical treatment of life-threatening conditions associated with AP, as well as development of minimally invasive techniques, which have enabled transperitoneal, retroperitoneal, transmural, or transpapillary approach to pancreatic necrosis [8,9,10,11,12,13,14].

The so-called ‘step-up approach’ is a widely accepted strategy for managing WOPN, including initial medical treatment of symptomatic necrosis with antibiotics and proper nutrition [8,10,11,12,15]. An indication for invasive treatment is persistent symptoms despite medical therapy [8,10,11,12,15]. According to the ‘step-up approach’, access should be gradually intensified using minimally invasive techniques; should these fail, the treatment of choice would be open necrosectomy [10,15]. In 2010, van Santvoort et al. demonstrated that gradually intensified minimally invasive treatment (i.e., the step-up approach) in pancreatic necrosis reduces the risk of complications, including lethal ones, compared to open necrosectomy [15]. In contrast, in 2017, van Bunschot et al.’s multicenter randomized trial did not show any difference in the risk for systemic complications in patients with pancreatic necrosis undergoing endoscopy (endoscopic step-up approach) compared to those treated with minimally invasive surgical techniques (surgical step-up approach) [16]. In the same study, the authors established that patients with pancreatic necrosis treated endoscopically developed fewer complications, such as pancreatic fistulas, and had shorter hospital stays compared to those treated surgically [16].

## 3. Transmural Endoscopic Drainage

In transmural endoscopic drainage of PFCs, the fluid is removed through a fistula created between the lumen and gastrointestinal tract (stomach or duodenum) [17,18,19,20,21]. The first reports on successful endoscopic drainage of post-inflammatory PFCs were published in the 1980s, and covered conventional drainage, where cystogatrostomy or cystoduodenostomy were created on top of the endoluminal bulge (luminal compression) posed by the PFC on the gastrointestinal wall [17]. The endoscopic technique for conventional endoscopic drainage limited the use of this method due to the fact that most PFCs are located within the tail, and do not compress the gastrointestinal wall. Furthermore, conventional endoscopic drainage was associated with a high risk of bleeding from blood vessels, which are not visible on endoscopy.

In 1992, Grimm et al. first described transmural drainage under endoscopic ultrasonography (EUS) guidance [18]. EUS-guided fistulostomy allows for visualization of the collection and surrounding tissues in real time [22,23,24]. The use of EUS for endoscopic transmural drainage broadened the indications for endoscopic therapy, by enabling drainage of PFCs that do not compress the gastrointestinal wall. In numerous studies conducted over decades, it has been shown that EUS-guided transmural drainage is more effective and safe compared to conventional drainage [19,20,21].

First publications on transmural drainage focused on pancreatic pseudocysts [17,18], which are post-inflammatory PFCs containing liquid content without tissue fragments [1,2,3,4]. In pancreatic pseudocysts, transmural stents are inserted through the fistula (passive transmural drainage), which is usually sufficient. All definitions were presented in Table 1.

Transmural drainage of WOPN ( Figure 3a–d and Figure 4a,b) was first described in 1996 by Baron et al. [26]. The efficacy of endoscopic drainage in WOPN (Figure 5a,b) is much lower compared to endoscopic pseudocyst treatment [27,28] due to tissue fragments contained within WOPN [1,2,3,4].

The previously mentioned study comparing classic drainage with transmural EUS-guided drainage was conducted in patients with pancreatic pseudocysts [19,20,21]. While the use of EUS increases the efficacy of endotherapy [19,20,21], it was not until 2015 when it was established that EUS guidance in transmural WOPN drainage significantly reduces complications, including gastrointestinal bleeding and, thus, makes the procedure safer [29].

Patients with WOPN often require a more aggressive approach. Although passive transmural drainage is effective in managing pancreatic pseudocysts, it is not sufficient in WOPN. Therefore, it is necessary to apply active transmural drainage, during which the necrosis is flushed with saline by a transmural nasocystic catheter. In the first article covering pancreatic necrosis endotherapy, Baron et al. presented the outcomes of successful endoscopic treatment in 11 patients, in whom 10Fr stents or 7Fr nasocystic catheters were inserted following cystogastrostomy or cystoduodenostomy in order to rinse the collection with saline (active transmural drainage) [26]. Papachristou et al.’s study on 53 patients undergoing endoscopic transmural drainage for WOPN demonstrated a success rate of 81%, with a complication rate of 21% [30]. Smoczyński et al. presented the results of endoscopic treatment in 112 WOPN patients, of which the long-term success rate on a two-year follow-up was 90.4%, with a complication rate of 25.9% [31].

In the above-listed publications, the single transluminal gateway technique (SGT) was usually applied [26,30,31]; however, this may be insufficient in some patients, especially those with infected necrosis, even when multiple stents and nasocystic catheters are inserted into a single fistula. When SGT proves unsuccessful, it is necessary to broaden the access to the necrosis by various minimally invasive techniques. An example is the combination of endoscopic drainage with percutaneous drainage [32,33,34]. Considering local expertise, interventional strategy with minimally invasive procedures can be applied by starting with percutaneous drainage; if this fails, broadening the access by endoscopic drainage is followed [32,33]. Conversely, endotherapy can establish the foundation of interventional WOPN management; when it is not sufficient, percutaneous drainage can be done in addition [34].

The recent progress in endotherapy for pancreatic necrosis has made it possible to gain access to the necrosis by implementing endoscopic techniques, as well as limit the use of other minimally invasive techniques [35]. In early reports on endoscopic therapy for WOPN, a transmural fistula 10–12 mm in diameter was created between the gastrointestinal lumen and the necrotic collection [26]. In later reports, the diameter was increased up to 20 mm, which improved the outcomes [36]. The greater the cystogastrostomy or cystoduodenostomy diameter, the easier the outflow of the necrotic contents [37,38]. Increasing the fistula diameter not only improved the results of endotherapy, but also enabled introduction of a fiberoscope into the necrosis, followed by endoscopic necrosectomy [36,38,39,40,41]. During the procedure, the fiberoscope is introduced into the necrotic collection through the fistula, and the necrotic tissues removed using various endoscopic tools [38,39,40,41]. Endoscopic necrosectomy improved the outcomes of endoscopic therapy for pancreatic necrosis [38,39,40,41,42,43].

Development of endoscopic treatment for WOPN was presented in an article by Papachristou et al., who started to widen the fistula diameter to 8 mm by introducing nasocystic catheters or stents [30]. Later, the fistula was widened up to 20 mm, and alongside drainage, necrosectomy was performed as well by introducing an extraction balloon or Dormia basket and removing the necrotic tissues [30]. In the last phase, the fistula was widened up to 20 mm, and a gastroscope was inserted into the lumen to perform endoscopic necrosectomy [30].

Broader access to necrotic tissues provides better conditions for drainage. With the development of pancreatic necrosis endotherapy, not only did the fistula diameter increase; the number of transmural fistulas increased as well. Varadarajulu et al. presented the multiple transluminal gateway technique (MTGT), in which multiple transmural fistulas are created to connect the gastrointestinal lumen with the WOPN [44]. The authors demonstrated that the use of multiple routes (2–3) for transmural access to the WOPN (MTGT) is more effective than SGT [44]. This is especially effective when managing multilocular necrotic collections [45].

The next step in improving endoscopic access to necrosis was presented by Mukai et al., when extensive necrotic collections can be reached through a single fistula (single transluminal gateway transcystic multiple drainage [SGTMD]), without an additional transmural access [46,47]. Multiple accesses through a single transmural fistula (SGTMD) not only improve the results of endoscopic pancreatic necrosis treatment [46,47,48,49], but also limit the use of other minimally invasive methods, such as percutaneous drainage [35].

The greater the cystogastrostomy or cystoduodenostomy diameter, the easier the outflow of the necrotic tissues through the fistula, and thus, the better the draining conditions [50]. This statement encouraged researchers to utilize self-expanding metallic stents (SEMSs) for PFC drainage [51]. Lumen-apposing metal stents (LAMSs) have greater diameter, which enables easier outflow of necrotic content during endoscopic drainage [52,53,54], shortens treatment, and increases endotherapy efficacy compared to traditionally-used plastic stents [55,56,57]. The greater diameter of SEMSs also makes endoscopic necrosectomy easier. However, due to higher complication rates, including stent migration, and higher costs relating to metallic stents [52,54,58,59], not every patient with WOPN requires such stents for pancreatic necrosis drainage [58,60,61]. SEMSs (Figure 6a,b) should be reserved for endoscopy therapy in patients with extensive WOPN containing poorly-liquefied necrotic tissues, in which endoscopic necrosectomy may be necessary in the next step.

In recent years, a couple of publications emerged in current literature that describe efficiency of the Hot AXIOS lumen-apposing stents (Figure 7a–e) in the treatment of pancreatic fluid collections [62,63,64,65]. This type of stent is a novel double-flanged, covered, self-expanding metal stent [62,63,64,65], which may be used in endotherapy of pancreatic necrosis without necessity to use other endoscopic tools during first endoscopic procedure (under performed cystogastrostomy or cystoduodenostomy).

As mentioned above, two types of transmural drainage are available: Multiple plastic double-pigtail stents or SEMSs. Plastic stents are usually double-pigtail stents (7–10Fr) in order to avoid migration. SEMSs are fully covered metal stents, that allow to maintain wide lumen of transmural fistula. Some of studies that compared transmural endoscopic drainage with use of plastic stents and SEMSs showed, that use of metal stents shortens time of endoscopic procedure [12], improving clinical success rate [12]. On the other hand, use of SEMSs during endoscopic drainage of WOPN increases the risk of complications of endotherapy [12]. There are no clear guidelines explaining what kind of stents should be used during transmural endoscopic drainage (Table 2) [12].

## 4. Transpapillary Endoscopic Drainage

Pancreatic duct disruption leads to the spilling out of pancreatic juice, and results from acute or chronic pancreatitis, cancer, abdominal trauma, or surgery [25,66,67,68,69]. In the course of ANP, pancreatic duct disruption can occur [65,66], which manifests as a contrast leak (extravasation) into the necrotic collection during endoscopic retrograde pancreatography (ERP) [25,67]. According to various reports in the literature, pancreatic duct disruption can be observed in 38–81% of patients with ANP [25,66].

Partial disruption of the pancreatic duct (Figure 8a,b) is diagnosed when the pancreatic duct fills with contrast distal to the disruption site [55]. Complete disruption (Figure 9a) describes contrast extravasation outside the duct without contrast filling the distal part of the main pancreatic duct [69]. In the course of ANP, partial disruption of the pancreatic duct is more common than complete disruption [25]. Additionally, it is possible to observe no contrast spilling out of the duct (no visible disruption of the pancreatic duct) on ERP in some patients following ANP [25].

Transpapillary drainage is an option for patients with symptomatic WOPN in whom it is impossible to perform endoscopic transmural drainage because the distance from the gastrointestinal wall to the PFC is too large (above 15 mm), and contrast spilling was observed on ERP through the pancreatic duct disruption to the WOPN [35,70,71]. Moreover, if the WOPN does not completely resolve in response to transmural drainage, and the pancreatic duct communicates with the collection during ERP, transpapillary drainage can be additionally applied in some patients [35].

In active transpapillary drainage (Figure 10a–d), the nasocystic catheter or transluminal stent is inserted through the duodenal papilla to the pancreatic duct, the distal part reaching the necrotic collection through the disruption site [70,71]. Then, the collection is flushed with saline through the drain [70,71,72]. Active transpapillary drainage as the only access to the necrotic collection is an effective endoscopic method when the necrosis is liquefied, and the collection is not too large [70,71]. Usually, it is used alongside transmural and percutaneous drainage as part of the multiple gateway technique [31,34]. Active transpapillary drainage as a single gateway technique is rarely reported in literature and remains controversial. A study from 2015 found that transpapillary drainage can be safe and effective in patients with WOPN when transmural drainage is impossible, and the necrotic collection communicates with the pancreatic duct [71]. The results of endotherapy have been shown to be better in patients with partial disruption compared to those with complete pancreatic duct disruption [71].

In passive transpapillary drainage (Figure 11a–d), the endoscopic stent is introduced into the pancreatic duct to provide physiological outflow of the pancreatic juice to the duodenum, which is crucial to duct healing [25,69]. However, there are currently no guidelines on pancreatic duct stenting in patients with WOPN; thus, the use of endoscopy in managing pancreatic duct disruption due to ANP remains unclear. Most studies investigating endotherapy for pancreatic duct disruption in the management of PFCs focused on pancreatic pseudocysts, with often contradictory results [73,74,75]. Trevino et al. showed that pancreatic duct stenting during transmural drainage increases the efficacy of endotherapy [73]. In contrast, Hookey et al. found no significant differences as to treatment effectiveness between patients with transmural drainage and those who were subjected to both transmural drainage and pancreatic duct stenting [74]. In the same study, higher relapse rate was observed in patients with transmural and transpapillary drainage compared to those with only transmural drainage [74]. A study by Yang et al. [75] obtained similar results to Hookey et al. [74]. The authors demonstrated that pancreatic duct stenting does not have any positive impact on the outcomes of patients with transmural PPC drainage; furthermore, it also has a negative effect on long-term endoscopic therapy outcomes for PFCs [75]. In contrast, a large study from 2018 (226 patients with WOPN) found that endotherapy for pancreatic duct disruption secondary to ANP is the key component of endoscopic therapy for WOPN [25]. Pancreatic duct stenting is an effective method for managing pancreatic duct disruption, improves long-term outcomes of endoscopic therapy in patients with WOPN, and reduces recurrent PFCs [25]. In the same study, endoscopic therapy was shown to be the most effective method in partial pancreatic duct disruption compared to complete disruption [25].

Another issue with pancreatography in patients with consequences of ANP is pancreatic fragmentation (also known as disconnected duct syndrome [DDS]), which is diagnosed in patients with pancreatic duct disruption or contrast-filled segment of the main pancreatic duct (MPD), without contrast flow outside the duct in ERP, who are also shown to have a distal part of the pancreas on other imaging examinations [25,67,68]. In the first report on endoscopic management of patients with DDS, endotherapy proved to be effective and safe [67]. However, later reports on endotherapy in patients with fragmented pancreas demonstrated a high failure rate of endoscopic treatment, high rate of recurrent PFCs, and a need for surgical treatment [25,76]. In the case of DDS, if the outflow from the main pancreatic duct is closed (success of endoscopic treatment of disruption of the pancreatic duct), the proximal (i.e., disconnected) fragment of the pancreas will continue to supply the pancreatic fluid collection; thus, the collection will recur when the transmural prosthesis is removed [25]. Therefore, in patients with DDS, passive transmural drainage is necessary, and a permanent transmural stent should be left in place to prevent collection recurrence [25].

## 5. Limitations of Endotherapy of Walled-Off Pancreatic Necrosis

The main limitation of endoscopic treatment of consequences of acute necrotizing pancreatitis is relatively long time of therapy, which is usually connected with the necessity of active drainage of pancreatic necrosis. However, time of drainage varies and depends on amount of liquefied necrotic tissues in the lumen of necrotic cavity, depending on time since ANP onset. Very frequently the long time of endotherapy is caused by need of multiple treatments. Limitations mentioned above very often prolong hospitalization’s time of the patients with pancreatic necrosis under endotherapy. Another limitation of endoscopic treatment is the low ability of the endoscopic instruments to perform the debridement of the necrosis or necrosectomy. Moreover, despite use of LAMSs during endotherapy the remains problem of the low ability to obtain a wide communication between the lumen of gastrointestinal tract and the lumen of WOPN. The greater the cystogastrostomy or cystoduodenostomy diameter, the easier the outflow of the necrotic tissues through the fistula, and thus, the better the draining conditions, which result in a shorter time of endotherapy [50].

As it has been mentioned above development of minimally invasive techniques enabled transperitoneal, retroperitoneal, transmural, or transpapillary approach to pancreatic necrosis [8,9,10,11,12,13,14]. Current literature gives access to manuscripts, which describe alternative minimally invasive methods of treatment of pancreatic necrosis [77,78,79,80], allowing to perform a one-shot treatment as a result of broad access to the necrotic cavity and a more extensive and safe debridement for the necrosis [77,78,79,80]. In order to demonstrate efficiency of mentioned methods further studies conducted on a larger number of patients are required.

## 6. Summary

Despite enormous progress in AP treatment over the past 30 years, managing the consequences of ANP remains controversial. While numerous publications on minimally invasive treatment for consequences of ANP are available, several aspects of interventional treatment, especially endoscopy, are still unclear in this population of patients. The present paper discussed these aspects and summarized the current knowledge on endoscopic therapy for consequences of ANP.

Endotherapy is an effective and safe minimally invasive treatment in patients with consequences of ANP. The evolution of endoscopic techniques presented in this article contributed to an increased success rate of endoscopic drainage, making endotherapy an alternative to other minimally invasive methods for pancreatic necrosis treatment. Designing an appropriate flushing system that would enable aggressive active drainage coupled with subsequent passive drainage is the foundation for successful WOPN therapy. Finally, the choice of route access should depend on the extent of the necrosis, as well as the experience of the medical center.

## Figures and Tables

**Figure 1 jcm-09-00117-f001:**
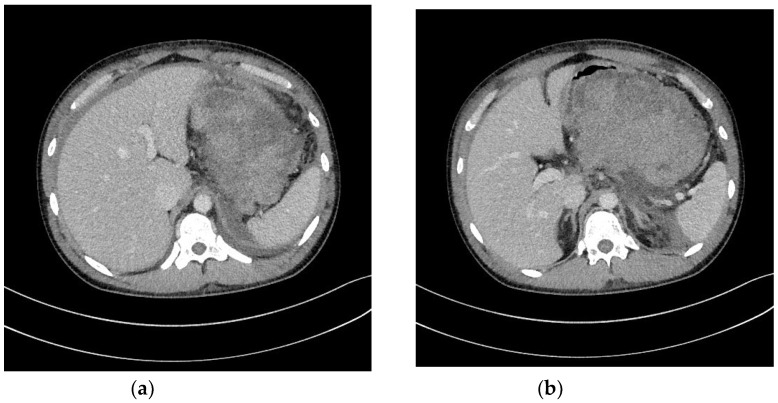
(**a**,**b**) Abdominal CECT was obtained in a 24-year-old male with ANP on day 9. In the pancreatic area, an ANC can be seen. ANC, acute necrotic collection; ANP, acute necrotizing pancreatitis; CECT, contrast-enhanced computed tomography.

**Figure 2 jcm-09-00117-f002:**
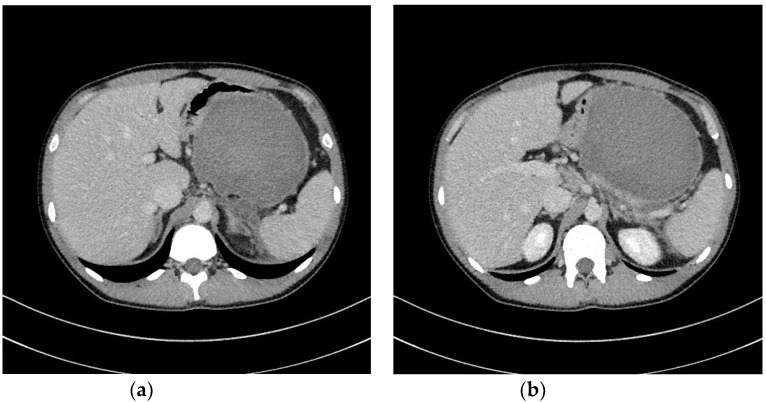
(**a**,**b**) Abdominal CECT was obtained in the same patient (Figure 1a,b) five weeks after the episode of ANP. A WOPN is visible, indenting the gastrointestinal wall. ANP, acute necrotizing pancreatitis; CECT, contrast-enhanced computed tomography; WOPN, walled-off pancreatic necrosis.

**Figure 3 jcm-09-00117-f003:**
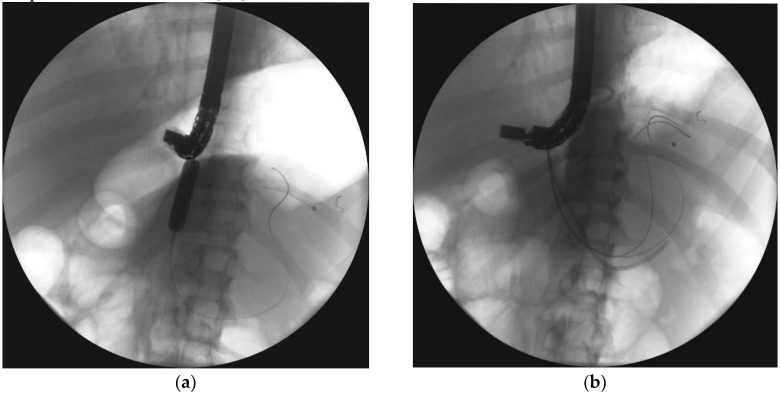
In the patient with symptomatic WOPN (Figure 2a,b), endoscopic transmural drainage was performed. Under EUS guidance, a cystogastrostomy was created. (**a**) The fistula was widened with a high-pressure 12-mm balloon. (**b**,**c**) Two stents and a nasocystic catheter were inserted into the necrotic collection. (**d**) Contrast agent was administered through the catheter and filled up the necrotic collection, which then drained freely into the stomach. EUS, endoscopic ultrasonography; WOPN, walled-off pancreatic necrosis.

**Figure 4 jcm-09-00117-f004:**
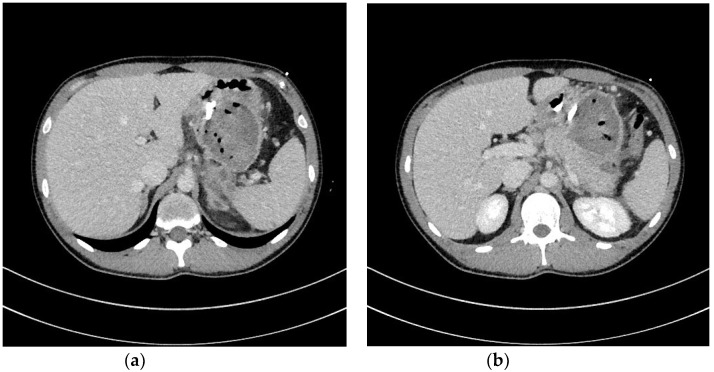
(**a**,**b**), Abdominal CECT was obtained during endoscopic transmural drainage (Figure 3a–d) of the WOPN. Transmural stents were inserted through the cystogastrostomy into the necrotic collection. CECT, contrast-enhanced computed tomography; WOPN, walled-off pancreatic necrosis.

**Figure 5 jcm-09-00117-f005:**
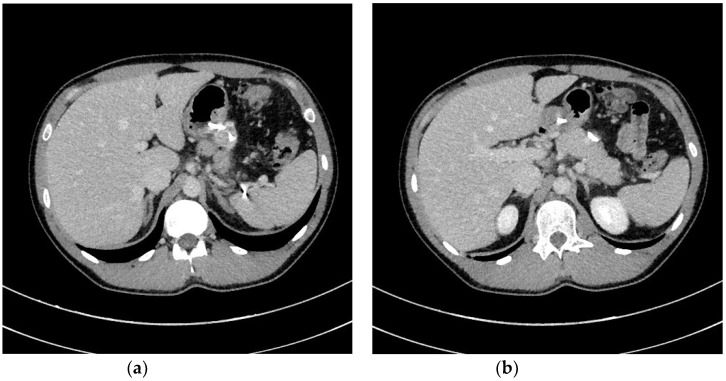
(**a**,**b**), Abdominal CECT was obtained after 36 days of active transmural drainage (Figure 4a,b). Complete regression of the necrotic collection can be appreciated. In the pancreatic area, transmural stents are visible. CECT, contrast-enhanced computed tomography.

**Figure 6 jcm-09-00117-f006:**
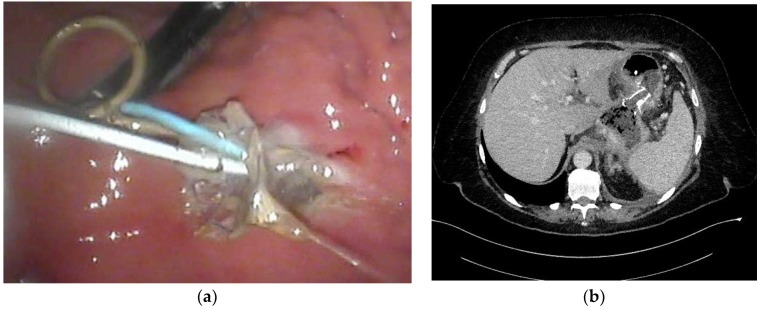
(**a**,**b**), SEMS in endoscopic transmural drainage of pancreatic necrosis. Two plastic stents and a nasocystic catheter were inserted into the WOPN through the metallic stent. Thick necrotic content is draining through the metal stent. SEMS, self-expanding metallic stent; WOPN, walled-off pancreatic necrosis.

**Figure 7 jcm-09-00117-f007:**
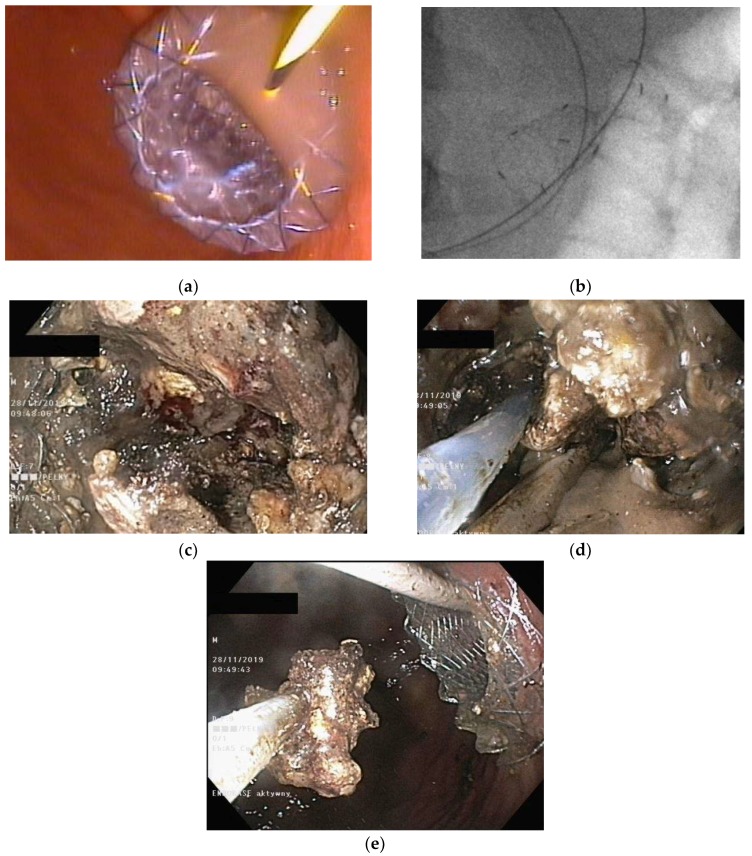
(**a**–**e**), Endoscopic treatment of WOPN with use of the Hot AXIOS lumen-apposing stents. During the first endoscopic procedure the cystogastrostomy was performed (**a**–**c**). The outflow of necrotic content through the transmural stent is visible. During the next endoscopic procedure, the endoscopic necrosectomy was performed. Through the lumen of stent, the gastroscope was inserted to the lumen of necrotic cavity and necrotic tissues were removed with the use of a Dormia basket.

**Figure 8 jcm-09-00117-f008:**
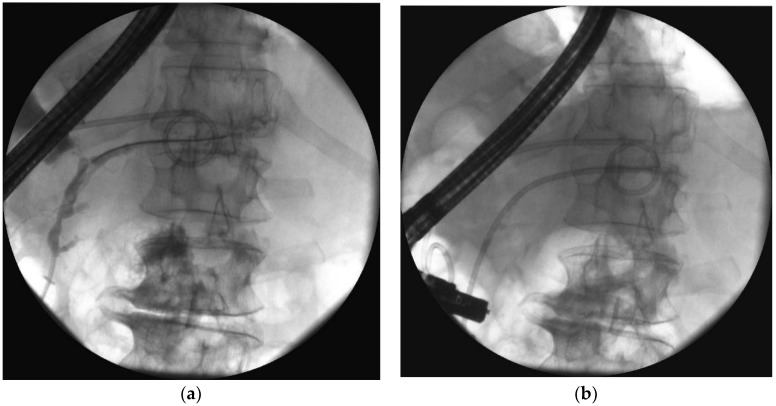
ERP during endoscopic treatment of pancreatic necrosis. (**a**) Complete disruption of the pancreatic duct can be seen within the tail at the level of transmural stent. (**b**) The stent bridging the pancreatic duct disruption. ERP, endoscopic retrograde pancreatography.

**Figure 9 jcm-09-00117-f009:**
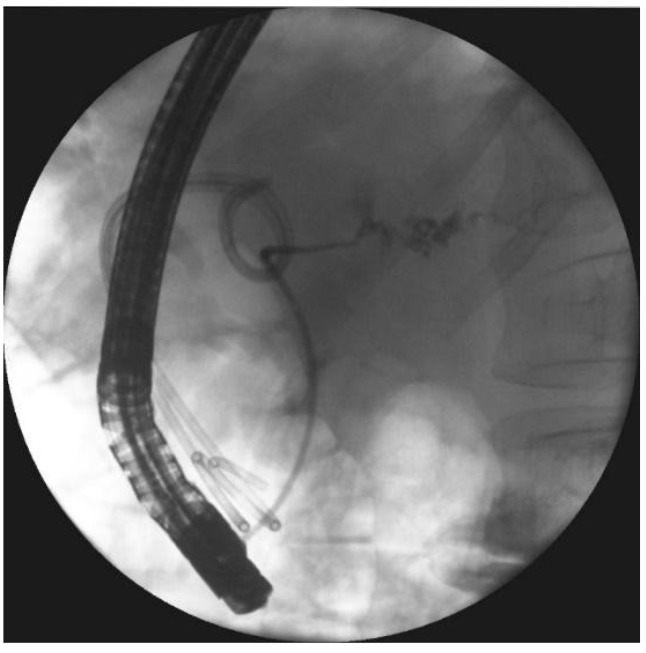
ERP was performed during transmural drainage of the WOPN. Complete disruption of the pancreatic duct within the tail can be seen, which causes contrast dye to spill outside the duct. ERP, endoscopic retrograde pancreatography; WOPN, walled-off pancreatic necrosis.

**Figure 10 jcm-09-00117-f010:**
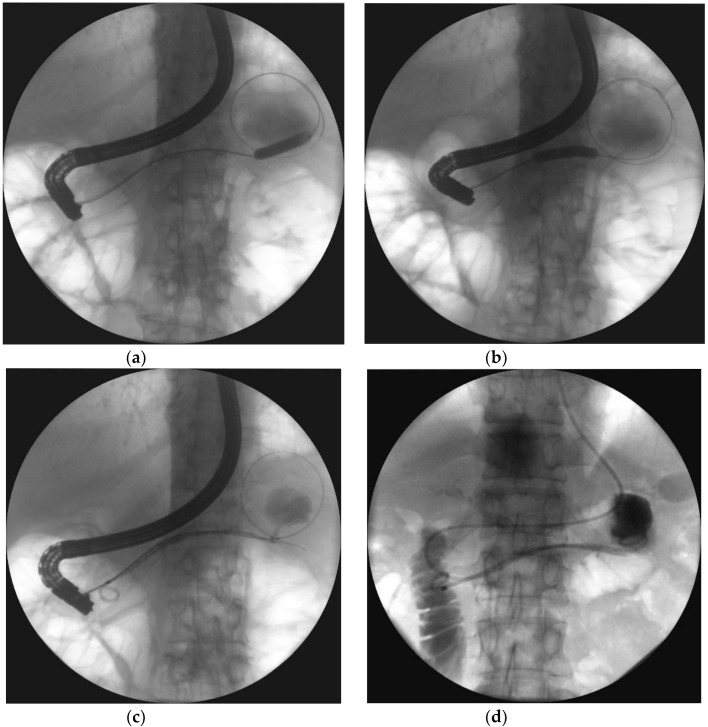
Endoscopic transpapillary drainage of symptomatic WOPN. (**a**,**b**) During ERP, the guide wire is introduced through the complete pancreatic duct disruption within the tail and looped within the central necrotic collection; the high-pressure 8-mm balloon is visible, which was used to dilate the pancreatic duct. (**c**,**d**) Transpapillary nasocystic catheter and stent insertion into the pancreatic duct, with the distal ends within the necrotic collection; contrast agent was administered through the catheter, filled up the collection, and drained freely into the duodenum.

**Figure 11 jcm-09-00117-f011:**
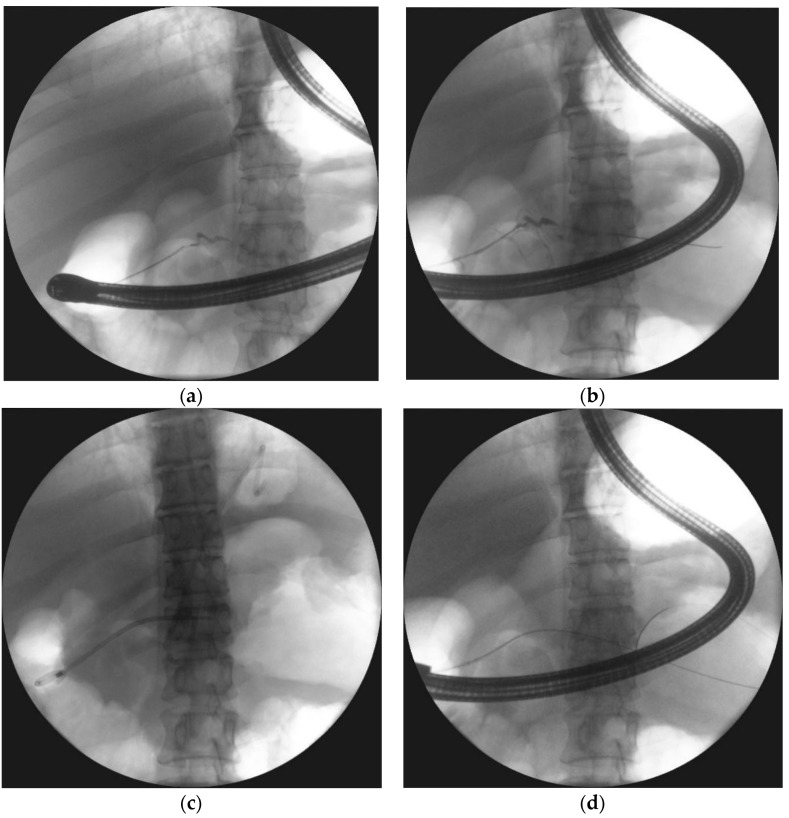
ERP was performed during endoscopic treatment for pancreatic necrosis. (**a**) Contrast dye is spilling out through the complete pancreatic duct disruption at the isthmus. (**b**,**c**) The guide wire was introduced through the complete pancreatic duct disruption and looped within the necrotic collection. (**d**) The stent was inserted through the duodenal papilla (red arrow) with the distal end within the collection, where the transmural stent is also visible ERP, endoscopic retrograde pancreatography.

**Table 1 jcm-09-00117-t001:** Definitions [25].

Term	Definition
Passive transmural drainage	The insertion of transmural endosthesis without the nasocystic drain in order to enable an outflow of necrotic content through the fistula and keeping the fistula unobstructed.
Active transmural drainage	The insertion of endoprostheses along with nasocystic drain through transmural fistula to the lumen of collection in order to enable WOPN flush.
Single transluminal gateway technique (SGT)	Complete removal of necrotic tissues through a single fistula created between the cavity of necrotic collection and the lumen of gastrointestinal tract (stomach or duodenum). Endoprosthesis and drains were inserted through the single fistula in the case of unilocular necrotic collections.
Multiple transluminal gateway technique (MTGT)	The creation of multiple transmural tracts between the gastrointestinal lumen and the WON cavity. In MTGT another transmural tract between the necrotic cavity and the gastrointestinal lumen was performed in case of multilocular necrotic collections divided by septa.
Single transluminal gateway transcystic multiple drainage (SGTMD)	Additional transmural drainage of extensive necrosis through a single fistula. Stents and nasocystic drains were introduced in the subcavities of WOPN through the single transmural tract and canals between necrotic subcavities.
Passive transpapillary drainage	The insertion endoscopic pancreatic stent to pancreatic duct through duodenal papilla.
Active transpapillary drainage	Endoscopic insertion pancreatic endoprosthesis as well as nasal drain to pancreatic duct through duodenal papilla.
The therapeutic success of WOPN	The absence of symptoms and complete regression of the necrotic collection.
The success of endotherapy of MPD disruption	The lack of contrast flow outside the MPD in patients with WOPN treated endoscopically for MPD disruption
Long-term success of WOPN treatment	The therapeutic success of WON endotherapy, the success of endotherapy of MPD disruption, and the lack of recurrence of PFCs during follow-up.

**Table 2 jcm-09-00117-t002:** Definitions [12,16,25,31,35].

Term	Definition and Indications
Transmural endoscopic drainage with use of plastic stents	Active (with nasal drain) or passive (without nasal drain) transmural drainage enabled via insertion of plastic stent or stents through the transmural fistula into lumen of necrotic cavity. Mostly used in cases of well-liquefied collections of WOPN with small number of necrotic tissues in the lumen of WOPN, which usually take place after six weeks from the beginning of ANP. This type of drainage should be used in cases, where there is no necessity to perform endoscopic necrosectomy.
Transmural endoscopic drainage with use of SEMSs	Active (with nasal drain) or passive (without nasal drain) transmural drainage accomplished via insertion of metal stent (SEMS) through the transmural fistula into lumen of necrotic collection. Indications for this type of drainage are extensive WOPN containing poorly-liquefied necrotic tissues, in which endoscopic necrosectomy may be necessary in the next step. SEMSs are usually used in the endoscopic treatment of WOPN up to sixth week from the beginning of ANP.
Endoscopic necrosectomy under fluoroscopic guidance (endoscopic debridement)	Procedure that enable to remove necrotic tissues from necrotic cavity through transmural fistula under fluoroscopy with use of various types of endoscopic tools. Indication for endoscopic debridement is WOPN containing poorly-liquefied necrotic tissues.
Direct endoscopic necrosectomy	Procedure accomplished via insertion of endoscope through the transmural fistula into the lumen of WOPN and direct removal of necrotic tissues under endoscopic view with use of different types of endoscopic tools. Direct endoscopic necrosectomy is usually technically easier to proceed during transmural drainage with use of SEMSs. Indications for direct endoscopic necrosectomy are extensive WOPN containing poorly-liquefied necrotic tissues without clinical improvement despite active transmural drainage.
Percutaneous drainage	Drainage enables to insert a drain transperitoneally or retroperitoneally into the lumen of necrotic cavity under control of ultrasonography or computed tomography and to flush the necrosis with saline solution through the percutaneous drain. This technique may be used as the only way to approach the necrosis or as additional approach (according to ‘step-up approach’ strategy).

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
