# Peer review of "Various Endoscopic Techniques for Treatment of Consequences of Acute Necrotizing Pancreatitis: Practical Updates for the Endoscopist"

_jcm, 2020, doi:10.3390/jcm9010117_

Round 1

Reviewer 1 Report

It’s a very interesting review about endoscopic techniques for treatment of consequences of acute necrotizing pancreatitis (ANP). Many aspects of endoscopic treatment of ANP have been well discussed and divided into categories: transmural endoscopic drainage and trans papillary endoscopic drainage. Indications and results were analyzed for these techniques. The figures are interesting and well represented.

This work is focused exclusively about the endoscopic treatment which is actually the most used in a step-up approach. However, I think that the limitations of this approach such as 1) the frequent need of multiple treatments; 2) the low ability of the endoscopic instruments to perform the debridement of the necrosis; 3) the low ability to obtain a wide communication between the stomach and the retro cavity, would deserve to be mentioned in the discussion. Furthermore, I think that alternative minimally invasive methods (e.g. laparoscopy and robot-assisted), more capable to perform a one-shot treatment because they allow a broad access to the necrotic cavity and a more extensive and safe debridement for the necrosis, should be cited (Gerin O, Prevot F, Dhahri A, Hakim S, Delcenserie R, Rebibo L, Regimbeau JM. Laparoscopy-assisted open cystogastrostomy and pancreatic debridement for necrotizing pancreatitis (with video). Surg Endosc. 2016 Mar;30(3):1235-41; Morelli L, Furbetta N, Gianardi D, Palmeri M, Di Franco G, Bianchini M, Stefanini G, Guadagni S, Di Candio G. Robot-assisted trans-gastric drainage and debridement of walled-off pancreatic necrosis using the EndoWrist stapler for the da Vinci Xi: A case report. World J Clin Cases. 2019 Jun 26;7(12):1461-1466), and discussed too.

Finally, in view of what said before, the statement " With an increasing number of patients with pancreatic necrosis, endotherapy may be the only mode of treatment " in the final section should be revised.

Author Response

Reviewer #1

It’s a very interesting review about endoscopic techniques for treatment of consequences of acute necrotizing pancreatitis (ANP). Many aspects of endoscopic treatment of ANP have been well discussed and divided into categories: transmural endoscopic drainage and trans papillary endoscopic drainage. Indications and results were analyzed for these techniques. The figures are interesting and well represented.

This work is focused exclusively about the endoscopic treatment which is actually the most used in a step-up approach. However, I think that the limitations of this approach such as 1) the frequent need of multiple treatments; 2) the low ability of the endoscopic instruments to perform the debridement of the necrosis; 3) the low ability to obtain a wide communication between the stomach and the retro cavity, would deserve to be mentioned in the discussion. Furthermore, I think that alternative minimally invasive methods (e.g. laparoscopy and robot-assisted), more capable to perform a one-shot treatment because they allow a broad access to the necrotic cavity and a more extensive and safe debridement for the necrosis, should be cited (Gerin O, Prevot F, Dhahri A, Hakim S, Delcenserie R, Rebibo L, Regimbeau JM. Laparoscopy-assisted open cystogastrostomy and pancreatic debridement for necrotizing pancreatitis (with video). Surg Endosc. 2016 Mar;30(3):1235-41; Morelli L, Furbetta N, Gianardi D, Palmeri M, Di Franco G, Bianchini M, Stefanini G, Guadagni S, Di Candio G. Robot-assisted trans-gastric drainage and debridement of walled-off pancreatic necrosis using the EndoWrist stapler for the da Vinci Xi: A case report. World J Clin Cases. 2019 Jun 26;7(12):1461-1466), and discussed too.

Finally, in view of what said before, the statement " With an increasing number of patients with pancreatic necrosis, endotherapy may be the only mode of treatment " in the final section should be revised.

Response:

Dear Reviewer,

In the beginning, I would like to thank you for a very positive revision of our manuscript.

We agree with the Reviewer and according to Reviewer’s suggestion, we have added a section: “Limitations of endotherapy of walled-off pancreatic necrosis”. In this section, we have thoroughly discussed all limitations of endoscopic treatment of pancreatic necrosis. Moreover, we have covered the issue of alternative minimally invasive methods by citing, among others, publications recommended by reviewer. In our opinion, additional section (“Limitations of endotherapy of walled-off pancreatic necrosis”) fully covers above-mentioned questions of the reviewer.

According to Reviewer’s recommendation, we’ve corrected the pointed out statement in the final section.

We hope that made corrections will satisfy both the Reviewers and the Editors.

We hope that our corrections will make the manuscript meet the requirements for publication in “Journal of Clinical Medicine”.

With kind regards,

Mateusz Jagielski

Reviewer 2 Report

This manuscript is a good review of the state of art about various endoscopic techniques for treatment of consequences of acute necrotizing pancreatitis.

Author Response

Reviewer #2

This manuscript is a good review of the state of art about various endoscopic techniques for treatment of consequences of acute necrotizing pancreatitis.

Response:

Dear Reviewer,

I would like to thank you very much for a very positive revision of our manuscript.

With kind regards,

Mateusz Jagielski

Reviewer 3 Report

Dear Authors,

I had the pleasure in reading your very well-organized manuscript. You might add a paragraph on the AXIOS (in the lumen opposing stents section) stents and could add a couple of photographs to elaborate this would be very good.

Best wishes.

Author Response

Reviewer #3

Dear Authors,

I had the pleasure in reading your very well-organized manuscript. You might add a paragraph on the AXIOS (in the lumen opposing stents section) stents and could add a couple of photographs to elaborate this would be very good.

Best wishes.

Response:

Dear Reviewer,

First of all, I would like to thank you for a very positive review of our manuscript.

We have made the appropriate corrections in the manuscript.

According to Reviewer’s suggestion, we have added a paragraph with description of  the AXIOS stents in the lumen opposing stents section. Moreover, according to Reviewer’s suggestion, we have added photographs (Figure 7 a-e) obtained during endoscopic drainage and endoscopic necrosectomy with use of this type of stents. We hope that added corrections increase the value of our manuscript.

We hope that made corrections will satisfy both the Reviewers and the Editors.

We hope that our corrections will make the manuscript meet the requirements for publication in “Journal of Clinical Medicine”.

With kind regards,

Mateusz Jagielski

Round 2

Reviewer 1 Report

The authors have addressed all the points raised by the reviewers.

Author Response

Reviewer #1

The authors have addressed all the points raised by the reviewers.

Response:

Dear Reviewer,

I would like to thank you very much for a very positive revision of our manuscript.

With kind regards,

Mateusz Jagielski
